# A Prospective Study on Deep Inspiration Breath Hold Thoracic Radiation Therapy Guided by Bronchoscopically Implanted Electromagnetic Transponders

**DOI:** 10.3390/cancers16081534

**Published:** 2024-04-17

**Authors:** Yuzhong Jeff Meng, Nikhil P. Mankuzhy, Mohit Chawla, Robert P. Lee, Ellen D. Yorke, Zhigang Zhang, Emily Gelb, Seng Boh Lim, John J. Cuaron, Abraham J. Wu, Charles B. Simone, Daphna Y. Gelblum, Dale Michael Lovelock, Wendy Harris, Andreas Rimner

**Affiliations:** 1Department of Radiation Oncology, Memorial Sloan Kettering Cancer Center, New York, NY 10021, USA; mengy1@mskcc.org (Y.J.M.); mankuzn@mskcc.org (N.P.M.); gelbe@mskcc.org (E.G.); cuaronj@mskcc.org (J.J.C.); wua@mskcc.org (A.J.W.); simonec1@mskcc.org (C.B.S.II); gelblumd@mskcc.org (D.Y.G.); 2Department of Medicine, Pulmonary Service, Section of Interventional Pulmonology, Memorial Sloan Kettering Cancer Center, New York, NY 10021, USA; chawlam1@mskcc.org (M.C.); leer2@mskcc.org (R.P.L.); 3Department of Medical Physics, Memorial Sloan Kettering Cancer Center, New York, NY 10021, USA; yorkee@mskcc.org (E.D.Y.); lims1@mskcc.org (S.B.L.); michael.lovelock@mountsinai.org (D.M.L.); harrisw@mskcc.org (W.H.); 4Department of Epidemiology & Biostatistics, Memorial Sloan Kettering Cancer Center, New York, NY 10021, USA; zhangz@mskcc.org; 5New York Proton Center, New York, NY 10035, USA; csimone@nyproton.com (C.B.S.II); 6Department of Radiation Oncology, Medical Center—University of Freiburg, Faculty of Medicine, University of Freiburg, German Cancer Consortium (DKTK), Partner Site DKTK-Freiburg, Robert-Koch-Strasse 3, 79106 Freiburg, Germany

**Keywords:** deep inspiration breath hold (DIBH), electromagnetic transponder (EMT), thoracic radiation therapy

## Abstract

**Simple Summary:**

We studied the feasibility and safety of using bronchoscopically implanted electromagnetic transponders to monitor deep inspiration breath hold (DIBH) for thoracic radiation therapy (RT) of primary lung cancers or lung metastases. Three transponders were implanted near the tumor, followed by CT simulation. The initial gating window was ±5 mm; in a second cohort, the window was incrementally reduced to determine the smallest feasible gating window: this was identified to be ±3 mm. Among the 48 patients enrolled, transponder-guided DIBH was feasible in all but two patients (96% feasible), where it failed because the distance between the transponders and the antenna was >19 cm. Toxicities at least possibly related to transponders or the implantation procedure were grade 2 in six patients, grade 3 in three patients, and grade 4 in one patient. Toxicities at least possibly related to RT were grade 2 in 18 patients and grade 3 in four patients. Transponder-guided DIBH is a feasible and safe approach for delivering thoracic RT.

**Abstract:**

Background: Electromagnetic transponders bronchoscopically implanted near the tumor can be used to monitor deep inspiration breath hold (DIBH) for thoracic radiation therapy (RT). The feasibility and safety of this approach require further study. Methods: We enrolled patients with primary lung cancer or lung metastases. Three transponders were implanted near the tumor, followed by simulation with DIBH, free breathing, and 4D-CT as backup. The initial gating window for treatment was ±5 mm; in a second cohort, the window was incrementally reduced to determine the smallest feasible gating window. The primary endpoint was feasibility, defined as completion of RT using transponder-guided DIBH. Patients were followed for assessment of transponder- and RT-related toxicity. Results: We enrolled 48 patients (35 with primary lung cancer and 13 with lung metastases). The median distance of transponders to tumor was 1.6 cm (IQR 0.6–2.8 cm). RT delivery ranged from 3 to 35 fractions. Transponder-guided DIBH was feasible in all but two patients (96% feasible), where it failed because the distance between the transponders and the antenna was >19 cm. Among the remaining 46 patients, 6 were treated prone to keep the transponders within 19 cm of the antenna, and 40 were treated supine. The smallest feasible gating window was identified as ±3 mm. Thirty-nine (85%) patients completed one year of follow-up. Toxicities at least possibly related to transponders or the implantation procedure were grade 2 in six patients (six incidences, cough and hemoptysis), grade 3 in three patients (five incidences, cough, dyspnea, pneumonia, and supraventricular tachycardia), and grade 4 pneumonia in one patient (occurring a few days after implantation but recovered fully and completed RT). Toxicities at least possibly related to RT were grade 2 in 18 patients (41 incidences, most commonly cough, fatigue, and pneumonitis) and grade 3 in four patients (seven incidences, most commonly pneumonia), and no patients had grade 4 or higher toxicity. Conclusions: Bronchoscopically implanted electromagnetic transponder–guided DIBH lung RT is feasible and safe, allowing for precise tumor targeting and reduced normal tissue exposure. Transponder–antenna distance was the most common challenge due to a limited antenna range, which could sometimes be circumvented by prone positioning.

## 1. Introduction

Radiation therapy (RT) is an effective treatment for patients with inoperable primary lung cancer and metastatic disease involving the lungs. High-dose stereotactic body radiation therapy (SBRT) results in excellent local control for early-stage non-small cell lung cancer (NSCLC) [1,2,3,4,5] and metastatic disease [6,7,8,9,10], and conventionally fractionated RT with concurrent chemotherapy is the standard of care in NSCLC patients who are medically inoperable or have unresectable locally advanced disease [11,12,13]. Toxicities of thoracic RT are related to the volume of normal tissue irradiated, tumor size and location, and dose [14,15,16,17,18,19]. A major challenge in thoracic RT is respiratory tumor and organ-at-risk (OAR) motion. Target volume delineation using a 4D simulation scan, with an internal target volume (ITV), accounts for respiratory motion and improves target coverage, but results in expanded target volumes and increased dose to lung parenchyma and other OARs, thus increasing the toxicity of RT.

In contrast to performing a free-breathing 4D simulation scan to account for motion, a reduction in respiratory motion during RT delivery can maintain target coverage and reduce the need for an ITV, resulting in a lower OAR dose. Such respiratory motion mitigation techniques include abdominal compression, respiratory gating, and breath-holding techniques [20]. Abdominal compression and respiratory gating have challenges with patient comfort and tolerance and inconsistency with anatomic variation [21,22,23,24,25,26]. Among the breath-holding techniques, deep inspiration breath hold (DIBH) has the advantage of expanding the lung volume to reduce the amount of lung tissue exposed to ionizing radiation and to increase the distance from targets to OARs [20,27,28,29,30]. Monitoring of DIBH can be achieved with spirometry, body surface surrogates, or internal surrogates. As spirometry and the movement of body surface surrogates do not correlate perfectly to movement of the tumor and can be subject to internal–external dissociation [23,24], internal surrogates hold great promise for the accurate monitoring of DIBH.

Implanted electromagnetic transponders (EMTs) in the lung are internal surrogates that allow active real-time tracking of tumor movement [31]. Its use has proven effective in prostate, liver, and pancreatic cancer at monitoring intrafraction motion [32,33]. A modified EMT was developed for bronchoscopic lung implantation and found to be safe and feasible [34,35,36,37,38,39]. We conducted an investigator-initiated prospective trial to study the use of EMTs to guide DIBH RT for primary and metastatic lung tumors. Previously, we reported that the EMTs were accurate and reproducible surrogates of tumor position [40]. Here, we report the primary endpoint of feasibility and safety outcomes of our study.

## 2. Materials and Methods

### 2.1. Eligibility

Patients were recruited, consented, and enrolled to this prospective investigator-initiated IRB-approved single-center feasibility study under a cross-referenced investigational device exemption (IDE) (NCT02111681). Enrolled patients had histologically confirmed primary lung cancer or lung metastases and were candidates for thoracic RT using DIBH. Treatments eligible for enrollment included stereotactic body RT (SBRT), hypofractionated RT, and conventionally fractionated RT. Eligibility criteria included the ability to perform a DIBH maneuver for >20 s five times in a row and candidacy for navigational bronchoscopy with airways in proximity to the target tumor amenable to endobronchial EMT implantation. Criteria for optimal airways for EMT implantation were an approximate diameter of 2 mm, >2 cm distance from the pleura, and <19 cm from the planned antenna/detector plate. Enrolled patients had a Karnofsky Performance Score (KPS) ≥ 60 and a life expectancy ≥ 12 months. Key exclusion criteria were metal implants in the chest region that could interfere with the electromagnetic localization, clinically active infections, bronchiectasis near the implantation site, hypersensitivity to nickel, and high risk for anesthesia or flexible bronchoscopy.

### 2.2. Procedures and Treatment

Enrolled patients underwent navigational bronchoscopy by a dedicated interventional pulmonologist (RPL and MC). Patients underwent standard pre-operative assessment prior to the procedure. During the procedure, three EMTs were implanted in small airways close to or inside the target tumor.

After EMT implantation, patients were observed for at least 3 days prior to simulation to allow for settling of any potential post-implantation EMT movement. All patients then underwent CT simulation with a DIBH scan and a free-breathing/4D-CT scan as backup in case they could not complete their entire treatment course with DIBH. All simulation scans were performed using respiration motion monitoring by external surrogates. The distance from the EMTs to the tumor was measured on the DIBH CT simulation scan from the closest point of the marker to the surface of the gross tumor volume (GTV).

Standard radiation dose, prescription criteria, and target and organs at risk volume delineation were used. Cone-beam CT images were obtained daily for SBRT and hypofractionated RT and weekly for conventional fractionation. During treatment, DIBH was monitored using an antenna positioned over the patient’s chest. Radiation therapy was delivered only when the centroid of the EMTs was within a gating window of their position on CT simulation, as confirmed by the day’s on-treatment imaging. For the first 25 patients, a gating window of ±5 mm was used. To determine the minimally clinically feasible gate margin, we then enrolled an expansion cohort and conducted a step-wise reduction in the gating window by 1 mm in each direction after each additional 5 patients. If a patient had substantial difficulty holding their breath within a gating window during the delivery of a particular treatment fraction, the gating window was increased for the remainder of that fraction. The duty cycle for every fraction was calculated and defined as the beam-on time as a fraction of total treatment time, where total treatment time does not include time for setup or imaging. If the gating window was changed during a fraction’s delivery, the beam-on time and total treatment time for that fraction included the delivery both before and after the change.

### 2.3. Follow-Up and Toxicity Assessment

Procedural complications from flexible bronchoscopy and adverse events related to anesthesia were recorded. During treatments, patients were assessed weekly for on-treatment toxicity. After treatment, patients were followed every 3 months for one year for protocol assessment of adverse events and with diagnostic CT chest imaging. Toxicities were assessed per Common Terminology Criteria for Adverse Events (CTCAE) v4.0. Adverse events were assessed for their relation to EMT implantation and to radiotherapy and were designated as unrelated, unlikely related, possibly related, probably related, or definitely related. After one year, patients continued routine follow-up. Clinical documentation and imaging studies were reviewed beyond the first year for assessment of local control of the treated lesion.

### 2.4. Statistical Analysis

The primary endpoint was feasibility of EMT-guided DIBH thoracic RT, which was defined as completion of all RT treatments using the DIBH treatment plan with monitoring by the implanted EMTs. We formed a 90% 2-sided Clopper–Pearson exact confidence interval for the feasibility rate. If the lower bound of this exact confidence interval was greater than 66%, then this approach would be declared feasible.

Time-to-event analysis was performed with the Kaplan–Meier method. The duration of local control was measured from the end of RT and censored at the last known CT scan.

## 3. Results

### 3.1. Patients

We screened 54 and enrolled 48 patients between 2014 and 2021, with a median age of 69. Among these patients, 35 had primary lung cancer and 13 had metastatic disease in the lungs. Detailed baseline characteristics are listed in Table 1.

### 3.2. EMTs Were Implanted Close to the Tumor with Low Toxicity

A total of 144 EMTs were implanted in the 48 enrolled patients. The median distance of EMTs to the surface of the GTV was 1.6 cm, with an interquartile range of 0.6–2.8 cm; 80.6% of all EMTs were within 3 cm of the GTV (Figure 1). One patient had one EMT migrate distally in the airway, as it was likely implanted in an airway too wide for implantation; they successfully completed EMT-guided DIBH and RT using the remaining two EMTs. The median time from bronchoscopy to CT simulation was 6 days, with an interquartile range of 4–7 days.

Only two instances of grade ≥ 2 toxicities occurred during or shortly after EMT implantation by navigational bronchoscopy under anesthesia (Table 2). One patient exhibited supraventricular tachycardia (grade 3) while under anesthesia and received esmolol, metoprolol, and amiodarone, after which the arrhythmia resolved; they were observed overnight and discharged the next day. Another patient had respiratory insufficiency immediately after the procedure, with chest X-ray showing worsening adjacent consolidation and atelectasis; they were managed with nebulizers and low-flow supplemental oxygen and discharged on the same day of the procedure. Six days later, they were admitted to an intensive care unit with post-obstructive pneumonia (grade 4) and septic shock; they recovered fully and completed RT.

### 3.3. Radiotherapy with EMT-Guided DIBH Was Feasible and Safe

EMT-guided DIBH was feasible in 46 out of 48 patients (96% feasible; 90% confidence interval 0.875–0.993, with lower bound above predefined threshold of 0.66). This treatment approach failed in two patients because the distance between the EMTs and the antenna was >19 cm and thus the EMT signal could not be successfully detected by the antenna. Among the remaining 46 patients, 6 were treated prone to keep the EMTs within 19 cm of the antenna, and 40 were treated supine. Thirty-five patients received SBRT, four received hypofractionated RT (8 or 15 fractions), and seven received conventionally fractionated RT (Table 1).

Two patients stopped RT early after 30/33 and 31/33 fractions, respectively, for clinical reasons unrelated to the DIBH feasibility or toxicity from the EMTs, and they were included in the analysis.

Thirty-nine (85%) patients completed 12 months of follow-up for toxicity. Toxicities at least possibly related to the presence of EMTs were grade 2 in six patients (six incidences, cough and hemoptysis) and grade 3 in two patients (four incidences, cough, dyspnea, and pneumonia). Toxicities at least possibly related to RT were grade 2 in 18 patients (41 incidences, most commonly cough, fatigue, and pneumonitis) and grade 3 in four patients (seven incidences, most commonly pneumonia), and no patients had grade 4 or higher toxicity (Table 2).

### 3.4. Gating Window Reduction

The gating window for EMT-guided DIBH was reduced from ±5 mm to ±2 mm. One patient in the ±4 mm gating window cohort and one patient in the ±3 mm gating window cohort had their gating window increased to ±5 mm for a portion of their treatments. However, in the ±2 mm gating window cohort, four patients needed to have their gating window increased to ±3 mm for a portion of their treatments; one patient needed to have their gating window increased to ±3, 4, or 5 mm for their entire treatment; and one patient had no changes to the gating window. As a result of these gating window adjustments, patients in all four gating window cohorts received treatment without a clear difference in duty cycle (Figure 2). Given that all but one patient in the ±2 mm gating window cohort required an increase in the gating window to make the DIBH treatment feasible, the ±3 mm gating window was declared the smallest feasible gating window for EMT-guided DIBH treatments.

### 3.5. DIBH Reduced PTV and Lung Dose Compared to 4D-CT

For 34 patients for whom both DIBH and 4D-CT treatment plans were available, we compared the plans and found that DIBH resulted in a median reduction in planning target volume (PTV) of 39.4% (IQR, 23.1%–49.2%) (Figure 3a) and a median reduction in mean dose to lung (excluding GTV) of 33.2% (IQR, 20.1%–41.8%) (Figure 3b). In two patients, the DIBH plan had a slightly larger PTV than the 4D-CT plan; in a separate patient, the DIBH plan had a slightly higher mean lung (excluding GTV) dose than the 4D-CT plan.

### 3.6. Local Control

In this heterogeneous cohort, the local control rate for all patients who completed EMT-guided RT was 86% at 12 months and 75% at 24 months, with a median follow-up of 28.3 months among those without a local failure (Appendix A). Among the subset of patients who received EMT-guided SBRT for early-stage NSCLC or lung metastases, the local control rate was 89% at 12 months and 77% at 24 months (Appendix A).

## 4. Discussion

In this prospective, investigator-initiated, single-center trial, we found that the use of bronchoscopically implanted EMTs for guidance of DIBH during thoracic RT for lung cancer and lung metastases was feasible and safe. EMTs were successfully placed within 3 cm of the tumor in 80.1% of instances without significant complications. Patients completed treatment using EMT-guided DIBH with low rates of adverse events. Our findings demonstrate the utility of bronchoscopically implanted EMTs by expert dedicated interventional pulmonologists and their relevance as a key technologic advancement in improving the delivery of thoracic radiotherapy.

Implanted EMTs have been used in radiotherapy of multiple disease sites. In prostate cancer, multiple institutions have shown their feasibility and effectiveness in monitoring intrafraction motion, and they have augmented advances in treatment margins and optimal treatment delivery with image guidance [41,42,43,44,45,46]. Use of EMTs has also been shown to be feasible in upper abdominal tumors in the liver and pancreas that are susceptible to intrafraction respiratory motion, with the potential for aiding in dose escalation [33,47,48,49,50]. For lung tumors, use of a reliable internal surrogate for monitoring of intrafraction motion is arguably much more important given the more significant respiratory excursion and irregular diaphragmatic motion, especially in the lower half of the lung [51]. With value demonstrated in other disease locations, lung-specific EMTs that could be anchored into the small airways near the tumor were developed showing promising feasibility in SBRT and further fractionated treatment courses of lung radiotherapy [34,35,36,37,38,39,40,52,53,54,55].

The published experiences of anchored EMTs in lung RT primarily utilize free-breathing or respiratory gating [35,38,39,53,54]. Target delineation using a 4D-CT and the corresponding expansion of target volumes to account for respiratory motion can increase the irradiated normal tissue dose, which can result in more toxicities or challenges in achieving the prescription dose to the tumor. DIBH has advantages compared to other motion management techniques of abdominal compression or active breathing control. Jaccard et al. first reported the use of EMT-guided DIBH in four patients, with reduced planning target volume (PTV) and OAR doses compared to 4D-CT plans [37]. Our group previously demonstrated that EMTs are good surrogates for tumor motion and are reliable in DIBH treatment [40].

We found that EMT-guided DIBH was highly feasible. The primary challenge was keeping the EMTs within the 19 cm operational range of the antenna array, which was resolved in six patients by prone positioning. During treatment, the distance limitation often required placing the antenna array very close to the patient; great care was often needed to allow adequate space for deep inspiration and for the patient’s nose in supine positioning. Cheng et al. described one patient treated with DIBH in a prone position due to a posterolateral tumor which resulted in a greater distance from the tumor to the heart compared to a supine, free-breathing approach [54]. However, they had another patient in whom two transponders were just greater than 20 cm from the detector and had a tracking failure. Sarkar et al. measured the depth of the tumor to the body surface prior to implantation [53]. Boggs et al. described identifying the most anterior skin location from the head to the lower abdomen as the position of the detector, as well as the placement of markers anterior to posteriorly located tumors. In their series, 3 of 16 patients had a challenge with collision and they used adjustments to immobilization, head rotation, and abdominal compression to successfully treat with transponder-guided gating [38]. Overall, all these factors should be considered in selecting appropriate candidates for transponder-guided RT. Extending the detection range of the antenna should be a priority in making this treatment technique feasible for more patients, especially patients with barrel chests from chronic obstructive pulmonary disease.

Essential to EMT-guided treatment is the safe and accurate placement of EMTs and the stability of their position. As previously discussed, variations in breathing pattern and tumor size during treatment confer the need for a surrogate near the tumor [34,56,57]. In our study, 80.1% of EMTs were placed within 3 cm of the GTV. Only one EMT migrated distally due to being implanted in a wide airway. We did not observe EMT migration during post-treatment follow-up. The largest published series of 69 patients reported 84.5% and 94.2% of transponders being placed within 3 cm and 4 cm of the tumor, respectively [39]. Sarkar et al. reported the migration of one EMT in a patient who developed a pneumothorax; the patient completed RT using the two remaining EMTs [53]. Schmitt et al. described a reduction in transponder distance due to tissue reaction and tumor shrinkage during hypofractionated and conventionally fractionated RT [55]. However, others have shown that transponders remain stable throughout SBRT and conventionally fractionated treatment [37,40]. McDonald et al. compared anchored transponders with other commercially available lung fiducial markers and found that all anchored transponders were within 5 mm of their position at simulation in 65% of patients at a median follow-up of 25.3 months, with no late toxicities attributable to anchored transponders and possible migration [34]. Dobelbower et al. reported one patient with a transponder that migrated to the pleural space three months after RT causing a bronchopleural fistula and abscess [39]. Overall, we and others have found that EMTs can be placed accurately near the tumor with little migration.

Importantly, we report few adverse events related to the bronchoscopic implantation procedure. Among the 48 patients, there were only two grade ≥ 2 adverse events: one patient with grade 3 arrhythmia and one with grade 4 pneumonia, both of whom recovered and successfully completed RT. No patient experienced a symptomatic pneumothorax, which is likely a result of the recommendation to avoid EMT implantation within 2 cm of pleural surfaces. Dobelbower et al. reported implantation-related serious adverse events in 7 of 69 patients, which included pneumonia (*n* = 3), pneumothorax (*n* = 2), and short cardiac arrest (*n* = 1) and hypotension (*n* = 1) related to anesthesia [39]. In another study of 31 patients, incidence of asymptomatic pneumothorax was rare at 6% [58]. One study described a patient with a very peripheral tumor who experienced a small pneumothorax and transponder migration [53]. However, other smaller series noted no procedural complications [35,37,52,54]. Given ours and others’ experience, we would recommend against transponder implantation in patients deemed particularly susceptible to severe infection. Furthermore, transponders should not be implanted within 2 cm of the pleural surface given the risk of pneumothorax described by other series.

In our series, the patient who had distal migration of an EMT implanted in a wide airway successfully completed EMT-guided DIBH with the two remaining EMTs, as the system can standardly work with only two EMTs. Dobelbower et al. also had one patient in which only two transponders were present at treatment, but detection and tracking was still feasible [39]. Booth et al. had three patients where one transponder was not tracked. In one patient, this was due to one transponder being outside the detection range, but in the other two patients, it was felt that two transponders provided a superior surrogate [52]. We also had two patients in whom two ipsilateral tumors were treated utilizing the same three EMTs, which is highly dependent on the anatomy and location of the tumors in relation to each other and the EMTs, but feasible in select circumstances. Cheng et al. also demonstrated the feasibility of treating more than one tumor using respiratory gating in two right lower lobe tumors [54].

We investigated the smallest feasible gating window by reducing it incrementally from ±5 mm to ±2 mm. We found that five out of the six patients with a starting gating window of ±2 mm had an increase in gating window for some or all their fractions. Thus, the lower limit for a feasible gating window was determined to be ±3 mm.

We found that transponder-guided DIBH resulted in a median PTV reduction of 39.4% and a median reduction in mean lung dose of 33.2% compared to 4D-CT plans, consistent with findings by Jaccard et al. (median PTV reduction of 31%) and Cheng et al. (mean PTV reduction of 47.2%, mean reduction in mean lung dose of 20.2%) [37,54].

We believe EMT-guided DIBH is most useful in two scenarios: (1) lower lobe tumors near the diaphragm, which move great distances with respiration, and (2) stage III patients with large tumor volumes and high OAR doses. For ultra-central tumors, EMT implantation may not be feasible since nearby central airways are wide. Further research is needed on whether target margins can be reduced when DIBH is guided by EMTs compared to spirometry or body surface surrogates. Alternative noninvasive tracking technologies such as high-frequency MR-linac scanning should also be explored [59,60].

## 5. Conclusions

Bronchoscopically implanted EMTs represent an important technological advance in thoracic radiotherapy that allows for an internal surrogate for real-time tracking of tumor movement. RT with EMT-guided DIBH was highly feasible. EMT–antenna distance was the most common challenge due to limited antenna range, which could sometimes be circumvented by prone positioning. Toxicities with implantation, EMTs, and RT were low. Further studies are warranted to determine whether implanted EMTs can reduce target margins.

## Figures and Tables

**Figure 1 cancers-16-01534-f001:**
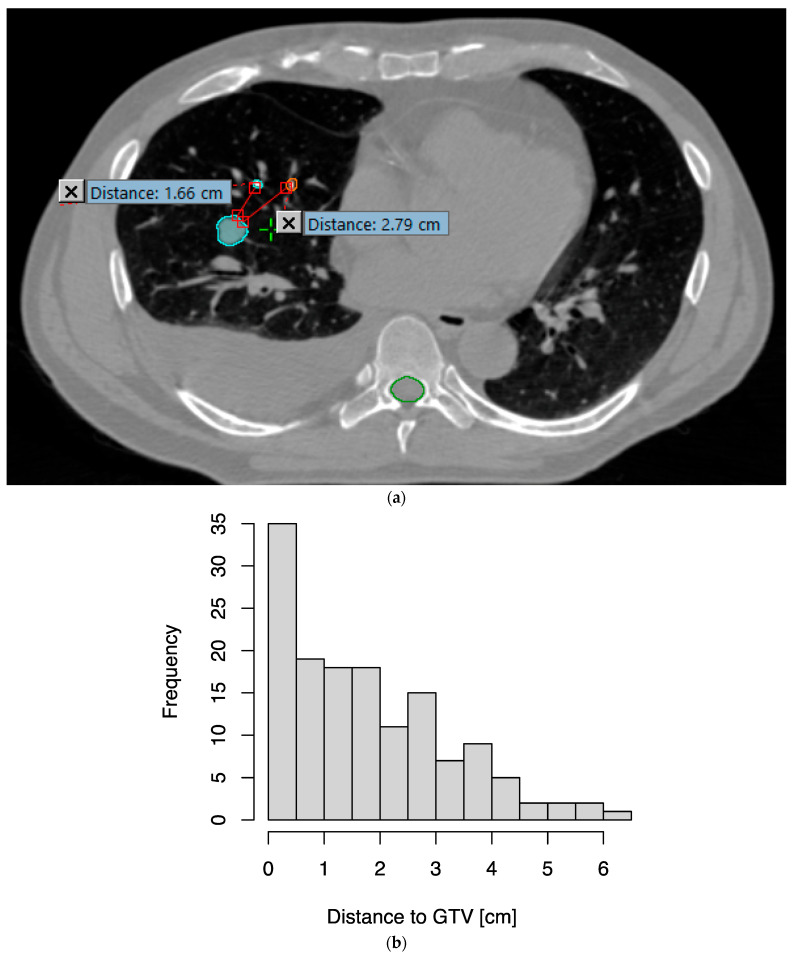
Distances of transponders from the gross tumor volume (GTV). (**a**) As an example, two of the three transponders implanted are seen in this axial slice from the simulation CT, at 1.66 cm and 2.79 cm from the surface of the GTV (cyan), respectively. (**b**) Histogram of distances from transponders to GTV.

**Figure 2 cancers-16-01534-f002:**
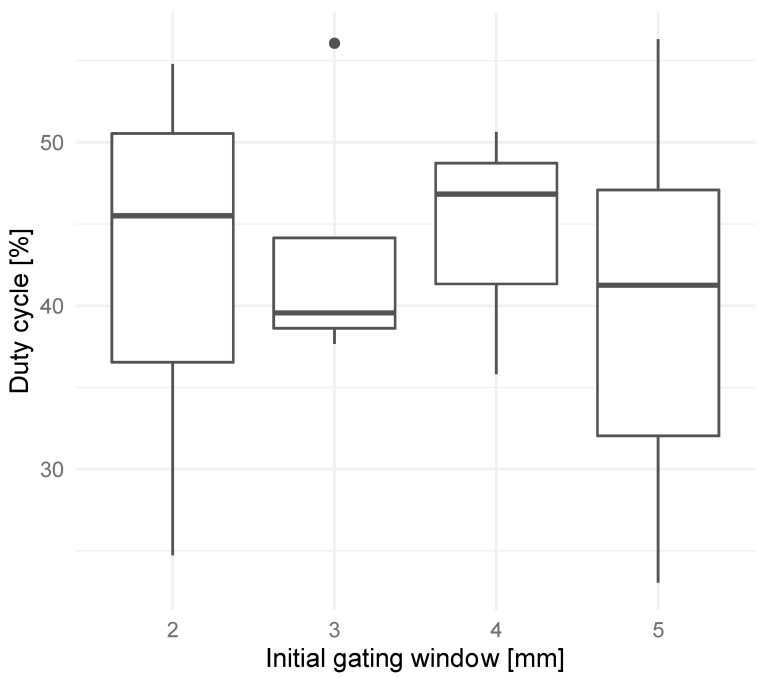
Mean duty cycles (percentage of beam-on time over total treatment time) over all treatment fractions for patients treated with initial gating windows of ±2, 3, 4, or 5 mm.

**Figure 3 cancers-16-01534-f003:**
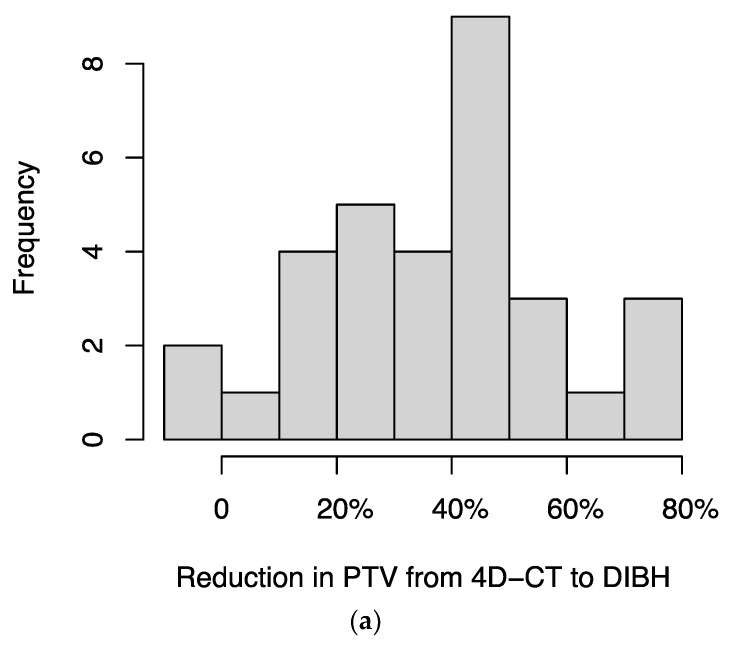
Histograms of the percentage reductions in (**a**) planning target volume (PTV) and (**b**) mean dose to lung (excluding GTV), comparing DIBH to 4D-CT plans for each patient. In two patients, the DIBH plan had a slightly larger PTV than the 4D-CT plan; in a separate patient, the DIBH plan had a slightly higher mean lung (excluding GTV) dose than the 4D-CT plan; these cases are represented by negative values on the x-axis.

**Table 1 cancers-16-01534-t001:** Patient and treatment characteristics.

Characteristic	*n* = 48
**Age**—Median (IQR)	69 (60, 73)
**Sex**—*n* (%)	
F	27 (56%)
M	21 (44%)
**Disease type**—*n* (%)	
Lung primary	35 (73%)
Lung metastasis	13 (27%)
**Stage**—*n* (%)	
I	15 (31%)
II	2 (4.2%)
III	6 (13%)
IV	22 (46%)
Local recurrence	3 (6.3%)
**RT type**—*n* (%)	
Stereotactic body RT	37 (77%)
Hypofractionated RT (8 or 15 fractions)	4 (8.3%)
Conventionally fractionated RT	7 (15%)
**Treatment position**—*n* (%)	
Supine	40 (83%)
Prone	6 (13%)
Not feasible	2 (4.2%)

**Table 2 cancers-16-01534-t002:** Incidences of toxicities (grade ≥ 2) related to implantation, transponders, or RT. Toxicities related to implantation occurred within two weeks of the procedure. Toxicities encountered in the first 12 months of post-RT follow-up were classified as unrelated, unlikely related, possibly related, probably related, or definitely related to transponders or to RT; all toxicities at least possibly related to implantation, transponders, or RT are counted here. When one patient experiences multiple toxicities of the same grade, the same patient is counted multiple times for the total number of incidences (the total number of patients are provided in parentheses). When one incidence of toxicity is at least possibly related to transponders and to RT, it is also counted in both categories.

Toxicities Related to Transponder Implantation			
**Grade**	**2**	**3**	**4**
Pneumothorax	0	0	0
Pneumonia	0	0	1
Supraventricular tachycardia	0	1	0
**Total**	0	1	1
**Toxicities Related to Presence of Transponders**			
**Grade**	**2**	**3**	**4**
Cough	5	1	0
Dyspnea	0	1	0
Pneumonia	0	2	0
Pulmonary hemorrhage or hemoptysis	1	0	0
**Total**	6 (6 pts)	4 (2 pts)	0
**Toxicities Related to RT**			
**Grade**	**2**	**3**	**4**
Cough	11	1	0
Dyspnea	2	1	0
Pneumonitis	6	1	0
Pneumonia	0	2	0
Chest wall pain	1	0	0
Dysphagia	4	1	0
Nausea	1	0	0
Vomiting	1	0	0
Anorexia	2	0	0
Dehydration	1	1	0
Fatigue	9	0	0
Dermatitis	2	0	0
Pulmonary hemorrhage or hemoptysis	1	0	0
**Total**	41 (18 pts)	7 (4 pts)	0

## Data Availability

Data are contained within the article and Appendix A.

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
