# Peer review of "A Prospective Study on Deep Inspiration Breath Hold Thoracic Radiation Therapy Guided by Bronchoscopically Implanted Electromagnetic Transponders"

_cancers, 2024, doi:10.3390/cancers16081534_

Round 1

Reviewer 1 Report

Comments and Suggestions for Authors

This study investigates the use of electromagnetic transponders, implanted near lung tumors via bronchoscopy, to monitor deep inspiration breath hold (DIBH) during thoracic radiation therapy (RT) for patients with primary lung cancer or lung metastases. The goal was to evaluate the feasibility and safety of this method. Forty-eight patients participated, with the transponders placed at a median distance of 1.6 cm from the tumors. RT was successfully delivered with DIBH guidance in 96% of cases. The optimal gating window for treatment was determined to be ± 3 mm. Transponder or implantation-related toxicities were generally low, with the most serious being a case of grade 4 pneumonia. The study concluded that using bronchoscopically implanted transponders for DIBH in lung RT is both feasible and safe. The primary challenge identified was maintaining the transponder-antenna distance within a usable range, occasionally addressed by treating patients in a prone position.

Minor comments:

1.     Even though calculations of overall survival are implied to be part of the study methods, it does not provide data on overall survival in the results section.

2.      Considering the fragility and common comorbidities of lung cancer patients, such as a history of smoking and chronic obstructive pulmonary disease (COPD), it would be valuable for the study to disclose how many patients were initially screened and subsequently excluded due to their inability to meet the criteria for deep inspiration breath hold. This information would provide further insight into the feasibility and applicability of electromagnetic transponder-guided DIBH for a broader patient population undergoing thoracic radiation therapy.

3.     The duty cycle which is reported in the study for each fraction of radiation therapy was calculated and defined as the beam-on time as a fraction of the total treatment time. Reporting the total treatment time alongside the duty cycle would be interesting as it provides a fuller picture of the treatment's practical implication.

4.     The study concludes by highlighting the need for further research on whether target margins can be reduced when using DIBH guided by electromagnetic transponders compared to traditional methods like spirometry or body surface surrogates. Additionally, I would suggest that MR-guided SBRT should be considered in this context, as it offers a noninvasive gating method that is less susceptible to the dissociation between internal tumor movement and external body surface movement.

5.     Limitations of the study should be discussed. E.g.: -The study did not report dosimetric data and differences in PTV and OAR compared to ITV-RT (although 4D-CTs were available). -Identifying a patient cohort that benefits from EMT implantation for thoracic RT remains an unmet need. -Due to the invasive nature of EMT implantation it might be considered, especially in situations where SBRT can otherwise not be delivered (e.g. ultracentral lung lesions)

Author Response

  1. Thank you for your comment about overall survival. In this heterogeneous cohort with patients spanning stage I to stage IV, we believe the overall survival is not a meaningful endpoint. Thus we removed the mention of assessing overall survival from our Methods, as you suggested.
  2. We agree with your comment that it would be helpful to tally the number of patients initially screened. We screened 54 patients and enrolled 48. We added this to Results.
  3. Thank you for your comment on examining treatment time. We did make a box plot of treatment time by gating window, and we found it to be dominated by differences in dose per fraction. Treatments with high dose per fraction (eg. SBRT) have longer treatment times, and SBRT patients were not uniformly distributed among the different gating window cohorts. Thus we found the analysis of treatment time by gating window to be less instructive than that of duty cycle by gating window.
  4. Thank you. We added MR linac to the end of the Discussion.
  5. Thank you for your comment. As you suggested, we examined difference in PTV volumes and lung doses between DIBH and 4D plans and added our analyses to the manuscript (Section 3.5 in Results, Figure 3, and parts in Discussion). We added to the last paragraph our thoughts on which patients may benefit the most from EMT-guided DIBH.

Reviewer 2 Report

Comments and Suggestions for Authors

Thank you for submitting this manuscript to Cancers. I read your work with great interest and congratulate your group on this work. 

The manuscript submitted investigates the feasibility and safety using transponder guided breath hold radiotherapy for lung cancer. This technique has been used in the management of other cancers previously but is a novel treatment approach for the management of lung cancers. The paper describes the bronchoscopic approach and details the issue of transponder migration and post procedure complication. I agree with the conclusion that this approach is safe and feasible and can lead potentially to reduced volumes of normal lung tissue exposed to high doses of radiation. The paper is well written and the nature of the study is such that a comparator cohort is not feasible. Comments on the Quality of English Language

Only very minor grammatical issues were detected.

For example, line 35 there is a significant gap before the number 5 .

Line 81 remove the word 'with'

Author Response

Thank you for your comments. We have addressed the grammatical and formatting issues you brought up.

Reviewer 3 Report

Comments and Suggestions for Authors

1.      Line 50: “Toxicities at least possibly related to RT were grade 2 in 41 patients (most commonly cough,fatigue, and pneumonitis) and grade 3 in 7 patients (most commonly pneumonia), and no patients had grade 4 or higher toxicity.” – all/most patients experience moderate (grade 2;  89%) or severe (grade3; 15%) toxicity – not sure in what standard that author could said “Bronchoscopically implanted electromagnetic transponder–guided DIBH lung RT is safe” (line 52). To me, this therapy is not safe.

2.      Effectiveness of any cancer treatment should be focused the local control rate and survival rate. Not sure why the author did not use a multivariable survival analysis model (hazard ratio model adjusted for important factors like cancer staging, age, different position (Supine vs Prone) etc.) to analyze the benefit of local control or risk of death after finishing RT treatment.

3.      Suggested to stratify patients with different cancer staging and generate two tables: after finishing this treatment, what were (1) local control rate (with 95% CI) and (2) survival rates (with 95% CI) in 3 months, 6 months, 1 year, 3 years and 5 years?

4.      Besides the overall graph – Figure S1 and line 233 – patients with “early-stage NSCLC or lung metastases” are very different – the survival graph lines should be stratified/separated by cancer staging groups.  Graph example: https://www.researchgate.net/figure/Kaplan-Meier-survival-curve-and-stages-of-cancer-among-prostate-cancer-patients-The_fig1_339729447

5.      Suggested to add a section mention the limitation of the study.

Author Response

1. Thank you for your comment on toxicity and safety. We find that the toxicities related to RT in our study were as expected for lung RT. You said that 89% of the patients experienced a grade 2 toxicity related to RT. We apologize for this misunderstanding. We counted incidences of toxicities. When one patient experiences multiple toxicities, the same patient is counted multiple times for the total number of incidences. To clarify, we also counted patients and amended our Abstract, Results, and Table 2. For example, “toxicities at least possibly related to RT were grade 2 in 18 patients (41 incidences…)” Furthermore, a grade 2 toxicity generally requires medical intervention but is not so severe to require hospital admission. Thus we believe the treatment is safe despite the number of grade 2 toxicities. Grade 3 or higher toxicities were uncommon.

2. Thank you for your comment on efficacy. We agree that local control and survival are good metrics of treatment efficacy. While we would love to investigate the influence of stage, age, and treatment position on treatment efficacy using a multivariable model as you suggested, we are unfortunately limited by the small sample size (n = 48) of our trial, which was powered for feasibility of EMT-guided DIBH.

3. Thank you for the suggestion to stratify local control and survival by stage. Again, we have a small sample size, and efficacy was not a focus of our study.

4. Same as above. For the reason you described (patients early stage NSCLC and lung metastases are very different populations), we focused our analysis on local control instead of overall survival.

5. Thank you for your suggestion. We added limitations of EMT-guided DIBH to the discussion.